# Physics of Discrete Impurities under the Framework of Device Simulations for Nanostructure Devices

**DOI:** 10.3390/ma11122559

**Published:** 2018-12-16

**Authors:** Nobuyuki Sano, Katsuhisa Yoshida, Chih-Wei Yao, Hiroshi Watanabe

**Affiliations:** 1Institute of Applied Physics, University of Tsukuba, Tsukuba, Ibaraki 305-8573, Japan; yoshida@bk.tsukuba.ac.jp; 2Department of Electrical and Computer Engineering, National Chiao Tung University, Hsinchu 30010, Taiwan; elegant.pegasus@gmail.com (C.-W.Y.); hwhpnabe@gmail.com (H.W.)

**Keywords:** random dopant, drift-diffusion, variability, device simulation, nanodevice, screening, Coulomb interaction

## Abstract

Localized impurities doped in the semiconductor substrate of nanostructure devices play an essential role in understanding and resolving transport and variability issues in device characteristics. Modeling discrete impurities under the framework of device simulations is, therefore, an urgent need for reliable prediction of device performance via device simulations. In the present paper, we discuss the details of the physics associated with localized impurities in nanostructure devices, which are inherent, yet nontrivial, to any device simulation schemes: The physical interpretation and the role of electrostatic Coulomb potential in device simulations are clarified. We then show that a naive introduction of localized impurities into the Poisson equation leads to a logical inconsistency within the framework of the drift-diffusion simulations. We describe a systematic methodology for how to treat the Coulomb potential consistently with both the Poisson and current-continuity (transport) equations. The methodology is extended to the case of nanostructure devices so that the effects of the interface between different materials are taken into account.

## 1. Introduction

Although device miniaturization by following the traditional scaling rule has already ended, the pursuit of the scaling merit of Si-based electron devices is now directed toward utilizing three-dimensional gate-surrounding structures of the channel substrate and/or replacing the channel material by a new material such as Ge or compound semiconductors. Even atomic layers such as MoS${}_{2}$ are also suggested as an alternative channel material [1]. Because of increasing complexity inherent to such advanced devices, the role of device simulation is getting more and more important [2]. In order to predict device characteristics accurately, it is essential to model physical phenomena based on the basic principles of physics. Local potential fluctuations induced by localized impurities, interface or line edge roughness, localized defects, etc., are just a few examples of such problems. Localized and, thus, discrete impurities doped in the device substrate induce surface potential fluctuations at the gate-oxide interface, which leads to threshold voltage fluctuations. This is called the random dopant fluctuations (RDFs) and a dominant factor that prevents further miniaturization of the present Si-based electron devices [3]. Intensive studies on the variability associated with discrete impurities have been, therefore, carried out in the past few decades [4,5,6,7,8,9,10,11,12,13,14,15,16,17]. The approaches employed in these studies scatter from the conventional drift-diffusion (DD) method to the Monte Carlo (MC) or the nonequilibrium Green’s functions (NEGF) methods [18,19,20,21,22,23]. Most simulations are, however, somewhat empirical; discrete impurities are introduced into the Poisson equation as point charges or by simply replacing the atoms of the substrate with charged ions, and the variability in device characteristics has been evaluated by brute-force means.

We would like to stress that the physical modeling of such potential fluctuations under the framework of device simulations is not trivial. An introduction of localized impurities into the device simulations implies a transition from the conventional continuous (long-wavelength limit) picture, which is a primary assumption of all device simulations mentioned above, to the discontinuous (discrete) picture. In other words, a naive introduction of point charges or similar ones into the Poisson equation may lead to a logical inconsistency with self-consistently-coupled transport equations [7,8,9,24]. Nevertheless, almost no attention has been paid to the physical aspects of such discrete impurities, except the present authors’ group. In the present paper, we thoroughly discuss the fundamental aspects of device modeling of randomly-doped discrete impurities within nanoscale device structures where the interface, as well as discreteness of impurities is of crucial importance. Although the physical issues are common to any kind of device simulations in which the Poisson equation is self-consistently coupled with the transport equations, we restrict our discussion here to the DD simulation scheme. A detailed analysis of the other simulation schemes along this line will be reported elsewhere.

The present paper is organized as follows. In Section 2, theoretical foundations imposed on the DD device simulations are discussed with emphasis on the length-scale involved in the scheme. In Section 3, the physics behind discrete impurities in the bulk is discussed, and a discrete impurity model appropriate for nanostructures where the interface effect between two different materials on the potential is inevitable is proposed. Finally, conclusions are drawn in Section 4.

## 2. Theoretical Foundations of Drift-Diffusion Device Simulations

The physical origin of RDFs in Si-MOSFETs has been properly recognized from the early stage of the investigations [7,8,9]; it is the long-range part of the Coulomb potential of doped impurities in the device substrate whose fluctuations lead to the variability of device characteristics. Since the potential fluctuations result from the discreteness of impurities, it is inevitable to introduce the discrete nature of impurities into the Poisson equation. Since the impurity density is described by a continuous and smooth function in the conventional scheme, this is not an easy task. If a point charge is introduced into the Poisson equation, the resulting Coulomb potential becomes so steep that carriers with opposite charge are trapped by the attractive potential (It should be noted that trapping and detrapping processes of carriers by ionized impurities are caused through the bound states created by the impurity Coulomb potential. These bound states, sometimes disguised in the DD simulations by the effective quantum potential by the density gradient method, are totally different from the free traveling states of carriers discussed here. This point should not be confused, as often seen in the literature, with the present issue.), and the doping density in the substrate is effectively lowered, which leads to an artificial threshold voltage shift [8,9]. Furthermore, as we shall discuss in Section 2.1, the short-range part of the Coulomb potential of impurities is double-counted because the conventional mobility model employed in DD simulations is dependent on impurity density, and thus, impurity scattering induced by the short-range (screened) Coulomb potential is already taken into account through the mobility model in the current-continuity equation [7,8,9,24,25]. A key to resolve this problem lies in the fact that we must treat the length-scales presumed in both the Poisson and the transport (current-continuity) equations in a consistent way.

### 2.1. Meaning of the Long-Wavelength Limit in Drift-Diffusion Simulations

The DD simulation scheme consists of the following Poisson equation,
(1)$$\begin{array}{c} \nabla \cdot \left[\varepsilon_{s}\nabla \phi \left(r,t\right)\right]=-e\left\{p\left(r,t\right)-n\left(r,t\right)+N_{d}^{+}\left(r\right)-N_{a}^{-}\left(r\right)\right\}, \end{array}$$
and the current-continuity equation for electrons,
(2)$$\begin{array}{c} \frac{\partial n\left(r,t\right)}{\partial t}-\frac{1}{e}\nabla \cdot J_{n}\left(r,t\right)=G_{n}\left(r,t\right)-R_{n}\left(r,t\right). \end{array}$$

Here, ϕ is the electric potential, *e* (>0) is the magnitude of the electron charge, *n* and *p* are electron and hole densities, $\varepsilon_{s}$ is the dielectric constant, $N_{d}^{+}$ and $N_{a}^{-}$ are ionized donor and acceptor densities, and $G_{n}$ and $R_{n}$ are generation and recombination rates per unit time. The current density of electrons is given by the sum of drift and diffusion current densities,
(3)$$\begin{array}{c} J_{n}\left(r,t\right)=en\left(r,t\right)\mu_{n}\left(-\nabla \phi \left(r,t\right)\right)-eD_{n}\left(-\nabla n\left(r,t\right)\right). \end{array}$$
where $\mu_{n}$ and $D_{n}$ are the mobility and diffusion constants of electrons, respectively. In addition, similar equations for holes are coupled with the above equations to determine the hole density. Notice that the current-continuity equation plays the role of the transport equation under the framework of the DD device simulations in the sense that the current-continuity equation determines carrier density (the Boltzmann transport equation and the Keldysh equation play the role of transport equations in the MC and NEGF device simulation schemes, respectively).

The Poisson equation given by Equation (Equation 1) holds true at any length-scale. Namely, if the charge density on the right-hand side is expressed in terms of the delta-functions (point charges), then the potential profile contains all wavelengths with no bounds. However, the potential usually assumed in Equation (Equation 1) is the one under the “long-wavelength limit” in the conventional device simulations so that the charge density of Equation (Equation 1) is expressed by a smooth continuous function (jelly impurity). This requirement is consistent with the current continuity Equation (Equation 3); the first term $\mu_{n}\left(-\nabla \phi \right)$ represents not the thermal velocity, but the drift velocity, which results from the collective (averaged) motion of electrons, and the second term yields the diffusion current, which results from the gradient of a smooth continuous electron density. Therefore, the DD simulation scheme is indeed consistent with respect to the length-scale as far as all physical variables are expressed with those under the long-wavelength limit.

The mathematical meaning of the “long-wavelength limit” is interpreted as follows. Let us consider the microscopic impurity density $N_{micro}$ expressed by the delta functions such that:(4)$$\begin{array}{c} N_{micro}\left(r\right)=\sum_{i=1}^{N_{imp}}\delta \left(r-R_{i}\right)=\frac{N_{imp}}{\Delta V}+\sum_{i=1}^{N_{imp}}\frac{1}{\Delta V}\sum_{k\neq 0}e^{ik\cdot \left(r-R_{i}\right)}, \end{array}$$
where $N_{imp}$ is the number of impurities included in a small volume ΔV around the position r and $R_{i}$ is the position of the $i^{\mathrm{th}}$ impurity. Then, the macroscopic (jelly) impurity density $\bar{N}\left(r\right)$, i.e., the density under the long-wavelength limit, is given by averaging $N_{micro}\left(r\right)$ over the small volume ΔV,
(5)$$\begin{array}{c} \bar{N}\left(r\right)=\frac{1}{\Delta V}\int_{\Delta V}d^{3}r\;N_{micro}\left(r\right)=\frac{N_{imp}}{\Delta V}. \end{array}$$

This implies that the long-wavelength limit of impurity density is equivalent to taking account of only the zero-Fourier component of the microscopic impurity density given by Equation (Equation 4). In other words, when impurity density is expressed as a smooth continuous function of position, the impurity density is considered to be locally flat, so that the electrostatic potential induced by the impurities is also locally flat because the number of impurities included in the region ΔV is virtually regarded as constant. This situation is schematically shown in Figure 1.

The question arises why non-zero Fourier components of discrete impurity density (equivalently, of electrostatic potential induced by each discrete impurity) could be ignored in the Poisson Equation (Equation 1). The answer lies in the fact that the mobility employed in Equation (Equation 3) is usually modeled as a function of impurity density in DD simulations, as already mentioned above. Since the mobility is determined by scattering, the impurity density dependence of mobility implies that non-zero Fourier components of each impurity potential are regarded as scattering potential and included in the conventional mobility model in Equation (Equation 3). This is the reason why non-zero Fourier components of the discrete impurity density and, thus, of the impurity potential are eliminated in the Poisson equation. Otherwise, these Fourier components would be double-counted.

### 2.2. Incomplete Screening of the Long-Range Part of the Coulomb Potential

We would like to stress, however, that all non-zero Fourier components of the Coulomb potential of each impurity are not actually treated as scattering potential. Impurity-limited mobility μ or scattering time τ appearing in the formula of mobility, $\mu =e\tau /m^{*}$ ($m^{*}$: effective mass), is, in most cases, calculated with the screened Yukawa potential. In other words, it is the short-range part of the Coulomb potential with the wavelength shorter than the screening length $\lambda_{c}$ that is treated as the scattering potential. The non-zero Fourier components with the wavelength larger than $\lambda_{c}$, which is hereafter denoted as the long-range part of the Coulomb potential, is assumed to be completely canceled by the induced charges of screening carriers and, thus, ignored in both the Poisson equation and the current-continuity equations.

It should be noted that the above scenario holds true if the carrier density is nearly equal to or above the average impurity density. The device is, however, operated under the extreme nonequilibrium conditions, and the charge neutrality is also broken in the subthreshold regimes near the gate interface where carrier density is very small. As a result, ionized impurities are not completely screened by carriers, and some portion of the long-range part of the Coulomb potential is left unscreened and appears as potential fluctuations (on the other hand, the impurities are “over-”screened in the inversion regimes by carriers whose density could be larger than the impurity density. In this case, however, carriers do not see the charge of impurities once the carrier density exceeds the impurity density, and thus, such “over-screening” does not induce any long-range potential fluctuations.). This unscreened portion of the potential always exists, no matter how large the device is. The reason why such potential fluctuations are not observable in large devices is because the variability of device characteristics associated with potential fluctuations is self-averaged due to large number of impurities in the substrate [26,27]. As the device shrinks, self-averaging is no longer strong enough to suppress the fluctuations and the variability in device properties such that RDF becomes significant. Therefore, the physical origin of RDF is indeed the long-range part of the Coulomb potential resulting from incomplete screening, as conjectured in the previous studies [7,8,9].

Furthermore, we should notice that as the volume of the channel substrate is small and/or complicated such as fin or surrounding-gate (nanowire) structures, the boundary (interface) greatly affects the spatial distribution of induced charges for screening in the semiconductor substrate. In other words, the effect of interfaces needs to be taken into account properly to extract the long-range part of the Coulomb potential.

## 3. Discrete Impurity Models for Drift-Diffusion Simulation

Following the arguments in Section 2, we need to somehow introduce the long-range part of the Coulomb potential associated with incomplete screening of discrete impurities. A critical issue is how we could include such potential fluctuations within the framework of the DD simulations. The hint is provided by recalling the role of the long-range part of the Coulomb potential. The most obvious example is plasma wave excitations in electron gas; electrons tend to screen the external (or extra) potential disturbance, yet because of the inertia of electrons, electron density spatially exceeds or gets under the proper value of the density, and this leads to plasma oscillations. This phenomena exactly corresponds to the dynamical version of the complete and incomplete screening situations described in Section 2.2. Since the plasma oscillation results from the collective motions of electrons, it is natural to include the long-range part of the Coulomb potential as the self-consistent Hartree potential in the Poisson equation, not as scattering in the mobility model of the current-continuity equation. Along with this idea, we have previously proposed a discrete impurity model for DD simulations [7], in which the charge density of each impurity in the Poisson equation is spread over the screening length so that the short-range part of the Coulomb potential is eliminated from the self-consistent Hartree potential (Technically, this may look similar to the cloud-in-cell method used in the MC simulations. However, the concept behind this process is very different. In the cloud-in-cell method, the size of charged particle is dependent on the mesh employed in the simulations, whereas the size of charged particle is fixed in the present method with the screening length and, thus, independent of the mesh.).

### 3.1. Discrete Impurity in the Bulk Structure

We notice that the long-range part of the Coulomb potential is just the potential (except the sign of charge polarity) generated by induced charges to screen ionized impurities. Since the self-consistent potential $\phi_{sc}$ is given by the sum of the external (impurity) potential $\phi_{ext}$ and the induced potential δϕ, the long-range part of the impurity Coulomb potential $\phi_{l}$, which results from the Poisson equation under the framework of DD simulations, is obtained from:(6)$$\begin{array}{c} \phi_{l}\left(r\right)=-\delta \phi \left(r\right)=-\left\{\phi_{sc}\left(r\right)-\phi_{ext}\left(r\right)\right\}. \end{array}$$

Transforming it into the Fourier q-space and using the fact that $\phi_{sc}\left(q\right)=\phi_{ext}\left(q\right)/\varepsilon \left(q\right)$ with the (relative) static dielectric function $\varepsilon \left(q\right)$, Equation (Equation 6) is expressed as:(7)$$\begin{array}{c} \phi_{l}\left(q\right)=\left(1-\frac{1}{\varepsilon \left(q\right)}\right)\phi_{ext}\left(q\right) \end{array}$$
and the corresponding charge distribution $\rho_{l}\left(q\right)$ is given by:(8)$$\begin{array}{c} \rho_{l}\left(q\right)=\varepsilon_{s}q^{2}\left(1-\frac{1}{\varepsilon \left(q\right)}\right)\phi_{ext}\left(q\right), \end{array}$$
where $\varepsilon \left(q\right)$ is calculated by the self-consistent field (random phase) approximation [28] and given by:(9)$$\begin{array}{c} \varepsilon \left(q\right)=1+\frac{{q_{c}}^{2}}{q^{2}}F\left(q\right). \end{array}$$

Here, $q_{c}$ is the inverse of the Debye screening length and given by $q_{c}=\sqrt{\bar{n}\left(r\right)e^{2}/\left(\varepsilon_{s}k_{B}T\right)}$ with average carrier density $\bar{n}\left(r\right)$ at equilibrium (The interpretation of $\bar{n}\left(r\right)$ requires some care. From the arguments in Section 2, it should be interpreted as the carrier density at equilibrium under the condition that charge neutrality is preserved. Here, charge neutrality simply means that the (macroscopic) carrier density is equal to the background (macroscopic) dopant density. Thus, $\bar{n}\left(r\right)$ should take the same value as that at the flat-band condition even if actual carrier density is different in the depletion or inversion regimes.), $k_{B}$ is the Boltzmann constant, and *T* is temperature. Assuming that the carrier distribution function is approximated by the classical Boltzmann statistics, $F\left(q\right)$ is given by:(10)Fq=−2m∗kBTℏqReZℏq22m∗kBT,
where $m^{*}$ is the effective mass of carriers in the semiconductor, *ℏ* is the Planck constant divided by 2π, and $Z\left(\alpha \right)$ is the plasma dispersion function defined by:(11)$$\begin{array}{c} Z\left(\alpha \right)=\lim_{\delta \to 0+}\frac{1}{\sqrt{\pi }}\int_{-\infty }^{\infty }d\xi \frac{e^{-\xi^{2}}}{\xi -\alpha -i\delta }. \end{array}$$

Since we are concerned with the long-range part of the potential, $F\left(q\right)$ is, as usual, set to unity, and the dielectric function could be approximated by the well-known Thomas–Fermi expression.

Consequently, the long-range part of the impurity potentials $\phi_{l}$ in both q- and r-spaces is given by:(12)$$\begin{array}{c} \phi_{l}\left(q\right)=\frac{e}{\varepsilon_{s}}\frac{{q_{c}}^{2}}{q^{2}\left(q^{2}+{q_{c}}^{2}\right)} \end{array}$$
and:(13)$$\begin{array}{c} \phi_{l}\left(r\right)=\frac{e}{4\pi \varepsilon_{s}}\left(\frac{1}{r}-\frac{e^{-q_{c}r}}{r}\right), \end{array}$$
respectively. As expected, $\phi_{l}\left(r\right)$ is indeed the difference between the bare Coulomb potential and the short-range Yukawa potential. The corresponding charge distributions $\rho_{l}$ in q- and r-spaces are given by:(14)$$\begin{array}{c} \rho_{l}\left(q\right)=e\frac{{q_{c}}^{2}}{q^{2}+{q_{c}}^{2}} \end{array}$$
and:(15)$$\begin{array}{c} \rho_{l}\left(r\right)=e\frac{{q_{c}}^{2}}{4\pi }\frac{e^{-q_{c}r}}{r}, \end{array}$$
respectively. Notice that it is this $\rho_{l}\left(r\right)$ that should replace the charge density expressed by a point charge of discrete impurity in the Poisson equation. Then, we are able to extract the long-range portion from the bare Coulomb potential. We would like to stress again that this long-range potential appears as potential fluctuation when the charge neutrality condition in the substrate is broken so that the screening by carriers is incomplete. In other words, the screening effect usually suppresses potential fluctuation, and thereby, incomplete screening causes potential fluctuation.

It is very interesting to compare Equations (Equation 14) and (Equation 15) with those we have previously proposed to extract the long-range portion of the impurity Coulomb potential [7,8,9,25]. Figure 2 shows the charge distributions (in both q- and r-spaces) and the (long-range) potentials of the three different expressions; the Yukawa-like expression given by Equation (Equation 15), the long-range expression given in [7,8,9], and the Gaussian expression in [25] employed as an alternative model to eliminate artificial oscillations showing up in the charge density by the previous long-range model. Explicit formulas of all three models are also shown in the inset of Figure 2. It is clear that the only difference among the models is how the short-range part of the Coulomb potential is eliminated: the short-range part is sharply cut at $q_{c}$ in the previous long-range model, whereas the present and Gaussian models gradually eliminate the short-range part. As a result, the long-range potentials are slightly different near the origin. Yet, all models properly approach the bare Coulomb potential as the distance from the impurity becomes much greater than $1/q_{c}$.

It should be noted that all expressions of the charge distribution hold true only in bulk structures because they assume that screening is not disturbed by the boundaries. This situation breaks down in nanostructures, in which impurities are surrounded by an interface and/or boundary.

### 3.2. Discrete Impurity Including the Effects of Interface

We now extend the above discrete impurity model to the case where an impurity is located near interface of two different materials so that the boundary could modulate the long-range part of the impurity potential. In order to extract the long-range part of the Coulomb potential, we take a similar methodology to the one we have employed in Section 3.1.

Let us consider two different materials with the relative permittivities $\varepsilon_{1}$ and $\varepsilon_{2}$ that are separated by an infinite plane interface. An ionized impurity is then embedded in the material with $\varepsilon_{1}$ at a distance *a* (>0) from the interface. The cylindrical coordinates whose origin coincides with the position of the impurity are employed, as shown in Figure 3.

The external (impurity) potential $\phi_{ext}$ that satisfies the boundary condition at the interface is expressed as:(16)$$\begin{array}{c} \phi_{ext}\left(\rho ,z\right)=\frac{e}{4\pi \varepsilon_{1}}\int_{0}^{\infty }dk\left[e^{-k\left|z\right|}+\alpha e^{-k\left(z+2a\right)}\right]J_{0}\left(k\rho \right) \end{array}$$
for z≥−a and:(17)$$\begin{array}{c} \phi_{ext}\left(\rho ,z\right)=\frac{e}{4\pi \varepsilon_{1}}\int_{0}^{\infty }dk\left(1+\alpha \right)e^{kz}J_{0}\left(k\rho \right) \end{array}$$
for z≤−a. Here, $\alpha =\left(\varepsilon_{1}-\varepsilon_{2}\right)/\left(\varepsilon_{1}+\varepsilon_{2}\right)$, and $J_{0}\left(x\right)$ is the zeroth order Bessel function. The Fourier transform of $\phi_{ext}\left(\rho ,z\right)$ is then given by:(18)$$\begin{array}{c} \phi_{ext}\left(q\right)=\frac{e}{\varepsilon_{1}}\frac{1}{{q_{⊥}}^{2}+{q_{z}}^{2}}\left(1+\alpha e^{-q_{⊥}a+iq_{z}a}\right)\equiv \frac{e}{\varepsilon_{1}}\frac{1}{q^{2}}\left\{1+\gamma \left(q\right)\right\}. \end{array}$$

Here, $q=\left(q_{⊥},q_{z}\right)$, where $q_{⊥}$ and $q_{z}$ are, respectively, wavenumbers normal to and along the *z*-axis so that $q^{2}=q_{⊥}^{2}+q_{z}^{2}$.

The static dielectric function $\varepsilon \left(q\right)$ under the self-consistent approximation becomes
(19)$$\begin{array}{c} \varepsilon \left(q\right)=1+\frac{{q_{c}}^{2}}{q^{2}}\left\{1+\gamma \left(q\right)\right\}F\left(q\right). \end{array}$$

Noting that $\left|\gamma \left(q\right)\right|=\left|\alpha e^{-q_{⊥}a+iq_{z}a}\right|<1$ and $0<F\left(q\right)\leq 1$, we employ the same approximation as in the bulk case; $\varepsilon \left(q\right)$ is approximated by the simple Thomas–Fermi expression. Hence, we can write the charge distribution induced by the impurity (with opposite charge polarity) as:(20)$$\begin{array}{c} \rho_{l}\left(q\right)=\varepsilon_{s}q^{2}\left(1-\frac{1}{\varepsilon \left(q\right)}\right)\phi_{ext}\left(q\right)=e\frac{{q_{c}}^{2}}{q^{2}+{q_{c}}^{2}}\left\{1+\gamma \left(q\right)\right\}. \end{array}$$

Transforming it into the r-space, we obtain the charge distribution $\rho_{l}\left(\rho ,\;z\right)$ by:(21)$$\begin{array}{c} \rho_{l}\left(\rho ,\;z\right)=e\frac{{q_{c}}^{2}}{4\pi }\int_{0}^{\infty }dq_{⊥}\frac{q_{⊥}}{\sqrt{{q_{⊥}}^{2}+{q_{c}}^{2}}}\left[e^{-\left|z\right|\sqrt{{q_{⊥}}^{2}+{q_{c}}^{2}}}+\alpha e^{-\left(z+a\right)\sqrt{{q_{⊥}}^{2}+{q_{c}}^{2}}-q_{⊥}a}\right]J_{0}\left(q_{⊥}\rho \right). \end{array}$$

Notice that this expression is valid only in the region of z≥−a.

Figure 4 shows the charge distributions given by Equation (Equation 21) along the z-axis (ρ=0), which are induced by the impurity at a distance *a* from the interface. The charge distributions of three different distances from the interface are shown for two different oxides in z<−a; SiO${}_{2}$ ($\varepsilon_{2}=3.9$) and HfO${}_{2}$ ($\varepsilon_{2}=25$). The semiconductor material in z>−a where the impurity resides is chosen to be Si ($\varepsilon_{1}=11.8$). In Figure 4, the charge distributions given by the first and second terms of the integrand in Equation (Equation 21) are also shown.

We notice that the first term in Equation (Equation 21) is identical to the charge distribution given by Equation (Equation 15). Therefore, this term represents the charge distribution induced by the impurity itself and should be interpreted similarly to the case of the bulk. The second term results from the polarization charge at the interface at z=−a and appears only after the discreteness of impurity is taken into account. Furthermore, depending on the magnitude of $\varepsilon_{2}$ compared with $\varepsilon_{1}$, the polarity of induced polarization charge changes: It is positive for SiO${}_{2}$, whereas it is negative for HfO${}_{2}$. As a result, the total charge distribution is more heavily affected near the interface. It should be pointed out that Equation (Equation 21) is derived by ignoring the metal surface on top of the oxide layer. Strictly speaking, this effect should be also taken into account if the oxide thickness is very small. This might be the case of SiO${}_{2}$. However, the effect is negligible for the case of HfO${}_{2}$ because of large permittivity (and large thickness usually employed in reality).

In order to properly take into account the polarization at the interface of discrete impurity in DD simulations, the following methodology is suggested. Since the first term in Equation (Equation 21) is identical to Equation (Equation 15), an impurity charge should spread similarly to the case of the bulk in accordance with Equation (Equation 15). It should be noted, however, that the charge distribution extending over the other side of the material (z<−a in the present case) should fold back to the semiconductor side (z>−a). Otherwise, the impurity density within the semiconductor substrate would not be conserved. This is exactly the procedure we have taken in the previous models when an impurity is located near interface so that some portion of its charge distribution spreads over the oxide. In addition, the correction charge distribution $\delta \rho_{l}$ associated with the polarization at the interface needs to be included. As a result, the impurity charge density near the interface for DD simulations should be given by:(22)$$\begin{array}{cc} \rho_{l}^{DD}\left(\rho ,\;z\right) & =\rho_{l}^{bulk}\left(\rho ,\;z\right)+\delta \rho_{l}\left(\rho ,\;z\right)\\ & =e\frac{{q_{c}}^{2}}{4\pi }\frac{e^{-q_{c}\sqrt{\rho^{2}+z^{2}}}}{\sqrt{\rho^{2}+z^{2}}}+e\frac{{q_{c}}^{2}}{4\pi }\alpha \int_{0}^{\infty }dq_{⊥}\frac{q_{⊥}}{\sqrt{{q_{⊥}}^{2}+{q_{c}}^{2}}}e^{-\left(z+a\right)\sqrt{{q_{⊥}}^{2}+{q_{c}}^{2}}-q_{⊥}a}J_{0}\left(q_{⊥}\rho \right) \end{array}$$
where $\rho_{l}^{bulk}$ is evaluated over the entire region, namely the charge distribution extended beyond the interface (z<−a) is folded back to the semiconductor substrate (z>−a) symmetrically with respect to the interface, whereas $\delta \rho_{l}$ is evaluated only in the semiconductor substrate (z>−a) (we are currently applying for a patent on a more tractable method, which could be properly implemented in the DD simulators, to extract the long-range potential of the discrete impurity near the interface). Clearly, Equation (Equation 22) coincides with the discrete impurity model in the bulk as the impurity position becomes very far from the interface (The present model is applicable to the region where impurities are intentionally doped so that the macroscopic dopant density is well defined in the device substrate. The case in which impurities are not intentionally doped is out of the scope of the present analysis.).

## 4. Conclusions

We have discussed the details of the physics associated with localized impurities in nanostructure devices. The physical interpretation and the role of the electrostatic Coulomb potential of localized impurities under the framework of device simulations have been clarified. We have shown that a naive introduction of localized impurities into the Poisson equation leads to a logical inconsistency within the scheme of the DD simulation. We have developed a systematic methodology for how to treat the Coulomb potential consistently with both the Poisson and the current-continuity (transport) equations. We have demonstrated that this method naturally leads to the concept of the long-range discrete impurity model we have proposed before. The method has been extended to the case of nanostructure devices in which the effects of the interface between different materials are taken into account.

Finally, we would like to point out that the present analysis is also closely related to the treatment of the Coulomb potential in any device simulation schemes. The long-wavelength limit is usually the common assumption in the Poisson equation of most device simulations, and thus, a similar careful analysis on logical consistency between the Poisson and the transport equations is required. This issue is under progress and will be reported elsewhere.

## Figures and Tables

**Figure 1 materials-11-02559-f001:**
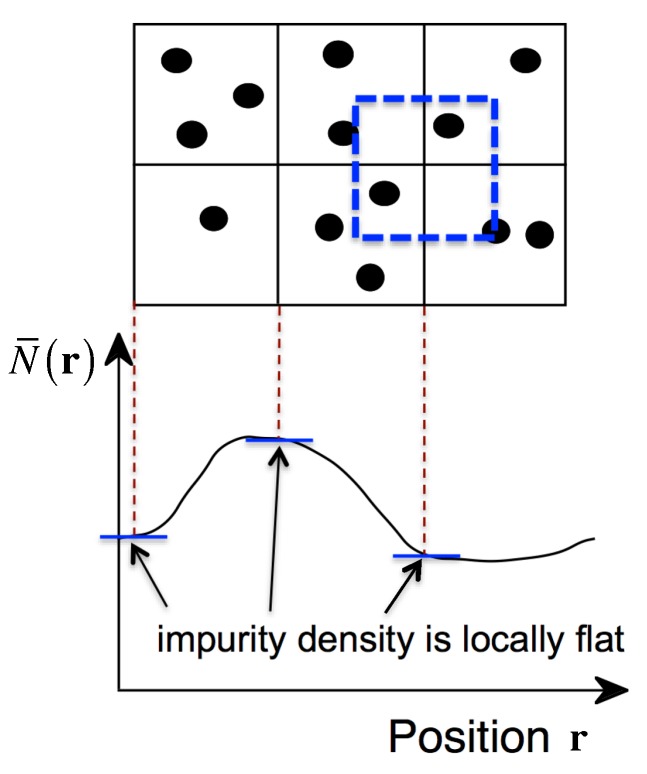
Schematic drawing of the spatial configuration of discrete impurities (above) and the corresponding macroscopic impurity density under the long-wavelength limit (below). The macroscopic density assumes that the number of impurities included in the neighborhood is virtually constant and the electrostatic potential is also virtually flat.

**Figure 2 materials-11-02559-f002:**
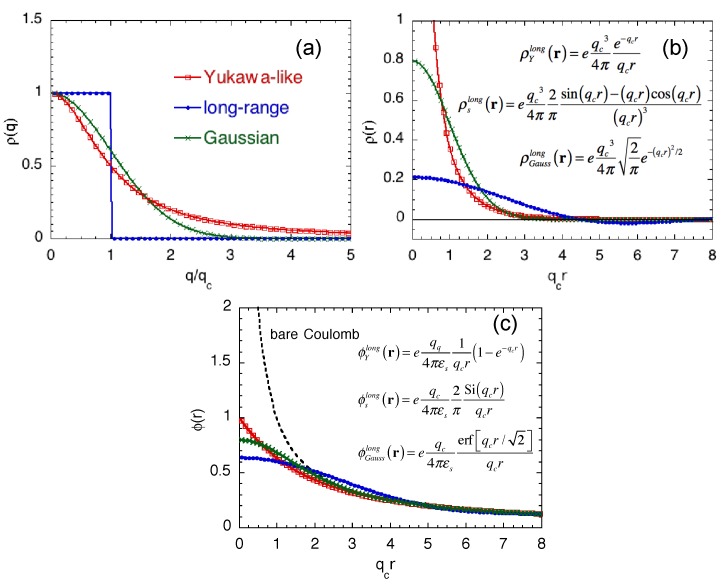
Charge densities of the discrete impurity located at origin as a function of (**a**) normalized wave-number $q/q_{c}$ and (**b**) normalized position $q_{c}r$ for three different discrete impurity models, Yukawa-like (red), long-range (blue), and Gaussian (green). The charge densities in q-space and r-space are, respectively, normalized by *e* and $eq_{c}^{3}/4\pi$. (**c**) Long-range potential of the three discrete impurity models as a function of normalized position $q_{c}r$. The bare Coulomb potential is also shown with the black dotted curve. The potential is normalized by $eq_{c}/4\pi \varepsilon_{s}$.

**Figure 3 materials-11-02559-f003:**
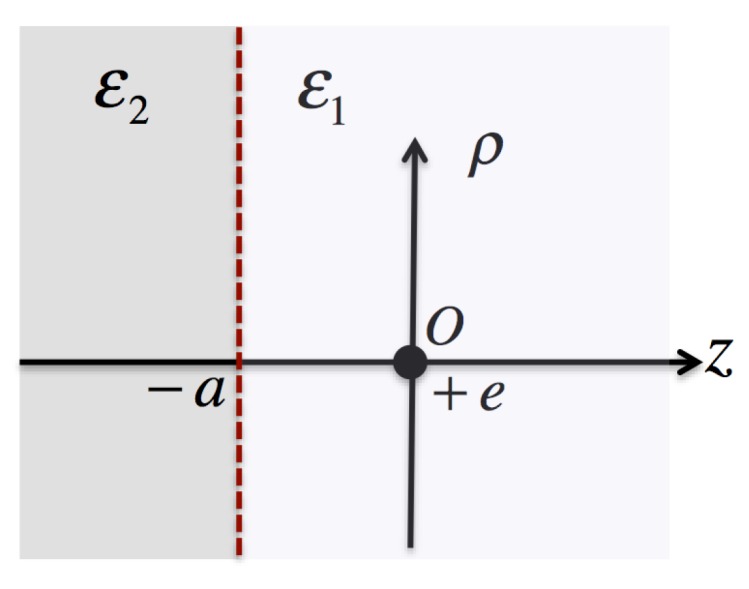
Schematic drawing of the interface between the two different materials with $\varepsilon_{1}$ and $\varepsilon_{2}$. A point-charge impurity is placed at the origin that is a distance *a* from the interface. The cylindrical coordinates are employed for the calculations.

**Figure 4 materials-11-02559-f004:**
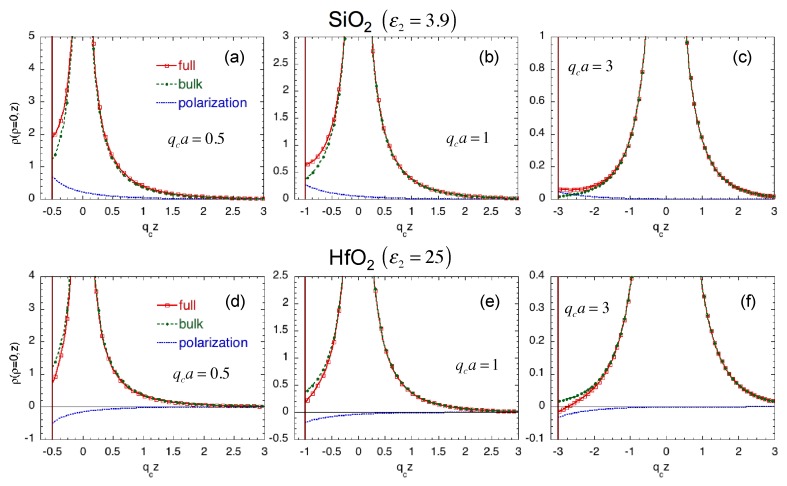
Charge distributions (denoted by “full” with red line) along the z-axis (ρ=0) induced by the impurity at the origin for two different dielectrics; SiO${}_{2}$ with $\varepsilon_{2}=3.9$ (**a**–**c**) and HfO${}_{2}$ with $\varepsilon_{2}=25$ (**d**–**f**). The distances from the interface are $q_{c}a=$ 0.5 (**a**,**d**), 1 (**b**,**e**), and 3 (**c**,**f**). The semiconductor material in z>−a is assumed to be Si ($\varepsilon_{1}=11.8$). The charge distributions obtained from the first term (denoted by “bulk” with the green line) and the second term (denoted by “polarization” with the blue line) in the integrand of Equation (Equation 21) are also shown.

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
