# Peer review of "Physics of Discrete Impurities under the Framework of Device Simulations for Nanostructure Devices"

_materials, 2018, doi:10.3390/ma11122559_

Round 1
Reviewer 1 Report
The present paper addresses the issue of discrete impurities in bulk and nanostructure devices, and proposes a new model for the long-range Coulomb potential and the impurity charge density. By approximating the dielectric function with the Thomas-Fermi expression, the Authors find that the long-range Coulomb potential is given by the difference between the bare Coulomb potential and the short-range screened Yukawa potential. From the Fourier transform of Poisson’s equation, they also infer the equivalent charge distribution associated with a single impurity atom, which turns out to be a function of the inverse Debye length. The analysis is then extended to the case of an impurity next to an interface between two different materials, such as a semiconductor and an oxide, both extending up to infinity. When the charge density extends beyond the interface, they suggest to fold back the distributed charge into the semiconductor, in order to keep the integrity of the total charge.
I have just a few critical remarks, reported below.
1) In footnote no. 4, it is stated that the Debye length and its inverse qc must be computed with “the carrier density at equilibrium under the condition that charge neutrality is preserved”. However, for a small number of discrete impurities within a nanoscale device, the concept of local charge neutrality becomes unclear. Next, the most important contribution to potential fluctuations is admittedly due to the existence of depleted regions, where a weaker screening effect takes place. In my view, the Debye length to be used in the proposed equations must be self-consistently computed with the local carrier concentration, even under non-equilibrium conditions.
2) The oxide thickness is typically smaller than the Debye length, especially for nanostructure devices, which are not intentionally doped in the active region. Therefore, the analysis of the discrete impurity next to the semiconductor-oxide interface should account for the metal surface on top of the oxide layer.
3) For nanoscale devices not intentionally doped within the active region, the local Debye length is orders of magnitude larger than the device itself. This poses the problem of charge conservation not just at the semiconductor-oxide interface, but also at metal contacts and at external boundaries of the simulated device.
4) The statement in the caption of Fig. 1: “The macroscopic density assumes that the number of impurities included in the neighborhood is constant and the electrostatic potential becomes flat” is incorrect. In numerical device simulations, the electrostatic potential is piecewise linear, not flat.
5) The statement: “the current-continuity equation plays the role of the transport equation in the framework of the DD device simulations” is incorrect. The DD transport equations are the constitutive equations of the semiconductor model based on Poisson and current continuity equations.
6) The statement “the long-range part of the impurity Coulomb potential ϕ to be included in the Poisson equation” is incorrect. Only the charge density is included in Poisson’s equation and the potential is the solution of the same equation.
7) The English of this manuscript requires some improvement. An excessive use of the article “the” should be corrected throughout the whole paper.
Minor points:
1) Line 32: “bruit-force means” should read “brute-force means”.
2) Caption of Fig. 4: “is assume” should read “is assumed”.
3) Line 14; “set unity” should read “set to unity”.
4) Line 42: “discrete impurities under nanoscale device structures” should read “discrete impurities within nanoscale device structures”.
5) Line 186: “the total charge distribution is greatly modulated near the interface” should read: “the total charge distribution is more heavily affected near the interface”.
In conclusion, this paper contains new and valuable information, which deserves to be conditionally published, after the above remarks are accounted for by the Authors.
Author Response
Reviewer #1
Thank you for many constructive comments to improve the manuscript. Our reply to each comment is given in the followings.
1) Charge neutrality simply implies that the (macroscopic) carrier density is equal to the background (macroscopic) dopant density. It does not matter whether or not “local” charge neutrality is actually preserved. To make this point clearer, we have added the statement in footnote 4, “Here, charge neutrality simply means that the (macroscopic) carrier density is equal to the background (macroscopic) dopant density.” (written in red in the revised manuscript).
About the second comment: As stressed in the manuscript, potential fluctuation results from “unscreened” portion of the Coulomb potential. This is possible only if the average carrier density equal to the background dopant density is used for qc. To make it clearer, the following statement has been added in p.6: “In other words, the screening effect usually suppresses potential fluctuations and, thereby, an incomplete screening causes potential fluctuations.”
2) Indeed, the effect of metal surface on top of the oxide layer should be taken into account if oxide thickness is very small. This may be the case of SiO2. On the other hand, the effect is negligible for the case of high-k materials because of large permittivity (and usually assumed) large thickness. This explanation has been added in p. 9.
3) The present model is applicable to the region where impurities are “intentionally” doped so that the macroscopic dopant density is well defined. The case in which impurities are not intentionally doped is out of scope from the present analysis. This statement has been added as footnote 7 in p.10.
4) As explained in the main text, the “macroscopic” density implies the density under the long-wavelength limit and Fig.1 illustrates the meaning of the “long-wavelength limit,” as stated in the caption. In order to make this point clearer, the word “virtually” was inserted in the caption of Fig. 1 and p.4.
5) All device simulations consist of transport equation and Poisson’s equation. Transport equation is the one that determines particle density: It is the BTE in the MC and the Keldysh equation in the NEGF, etc. In this sense, the current-continuity equation that determines carrier density plays the role of transport equation in the framework of the DD. This explanation was added in p.3 and footnote 2.
6) Indeed, this statement in p.5 is misleading. The statement was corrected as “which results from the Poisson equation under the framework of DD simulations”.
7) I have asked to check English of the manuscript and the manuscript was amended.
Thank you for pointing out typos and correcting the English expressions. They (and other typos) were corrected.
Reviewer 2 Report
1) There are many typos in the text which need to be corrected.
2) When citing the works [17-21], the authors could also cite the following interesting and appropriate work:
J.M. Sellier, I. Dimov,
The Wigner-Boltzmann Monte Carlo method applied to electron transport in the presence of a single dopant,
Computer Physics Communications, Vol. 185, pp. 2427-2435, (2014).
Author Response
Reviewer #2
Thank you for the comments to improve the manuscript.
1) We did our best to correct typos in the manuscript.
2) The reference suggested by the reviewer was added in the manuscript (as shown in red in the revised manuscript).
Reviewer 3 Report
The presented paper addresses an important issue of random dopant fluctuations in semiconductors. The research seems to be of growing relevance as the nowadays transistors are reaching dimension of 10nm and even 7nm in production. The paper is a theoretical analysis of the problem. I am not deeply familiar with the subject, so I cannot give an informed opinion on the derivations, but from a general perspective everything seems to be correct. The explanations are clearly written and understandable even for a reader outside of the field. I understand that there are already two reviews on this paper, hopefully from researchers more familiar with the subject, so the Editor should have enough insight on the technical quality of the presentation.
From my perspective, the only thing I can note is that I would have liked to see a more general starting paragraph in the Introduction. As it is now, the first sentence in the Introduction sounds like a part of a larger unsaid thought. For example, it is unclear what kind of “advanced devices” are considered, and what is meant by “traditional scaling merit”, and what are “channel materials”. A starting paragraph of a more general description and overview of the problem would be helpful.
From there on, the paper reads more consistently and things become clearer. The derivations are easy to follow and the authors give a lot of detailed explanations and side notes, which helps to better understand their reasoning.
Author Response
Reviewer #3
Many thanks for positive comments and also suggestions to improve the manuscript.
We have added more statements in the introduction to make Introduction clearer and more general (as shown in red in the revised manuscript).